# Polymorphism rs259983 of the Zinc Finger Protein 831 Gene Increases Risk of Superimposed Preeclampsia in Women with Gestational Diabetes Mellitus

**DOI:** 10.3390/ijms252011108

**Published:** 2024-10-16

**Authors:** Nataliia Karpova, Olga Dmitrenko, Malik Nurbekov

**Affiliations:** Federal State Scientific Institution “Research Institute of General Pathology and Pathophysiology,” the Russian Academy of Medical Sciences, 125315 Moscow, Russia; dolga6528@gmail.com (O.D.); mlkn0361@xmail.ru (M.N.)

**Keywords:** HDP, preeclampsia, superimposed preeclampsia, hypertension, CHTN, pregnancy, ZNF831, rs259983, gestational diabetes mellitus

## Abstract

Hypertensive disorders of pregnancy (HDP) are a great danger. A previous GWAS found a relationship between rs259983 of the ZNF831 gene and HDP, such as for chronic hypertension (CHTN) and preeclampsia (PE). We conducted the case-control study to determine the association between rs259983 of the ZNF831 gene and HDP in women with Gestational Diabetes Mellitus (GDM). For target genotyping, we developed primers and TaqMan probes. In analyzing the population, we did not manage to find a relationship between PE and rs259983 of the ZNF831 gene. Additional study of women with PE and PE superimposed on CHTN (SIPE) establishes an association between rs259983 of the ZNF831 gene only with SIPE. Carriers of CC genotypes have been discovered to have a 5.05 times higher risk of SIPE development in women with GDM.

## 1. Introduction

Maternal and neonatal health during and after pregnancy is considered the world’s biggest reproductive health concern [1]. According to a new United Nations report, there are about 4.5 million maternal and newborn deaths per year, the majority of which are preventable. HDP occurs in 5–10% of cases. It is one of the leading causes of maternal mortality and in 20–25% of cases of perinatal mortality [2].

Preeclampsia is one of the most serious HDP complications, occurring after 20 weeks of pregnancy and characterized by arterial hypertension in combination with proteinuria (≥0.3 g/L in 24 h urine), edema, and multiple organ dysfunction [3]. The prevalence of PE varies from 2.3% to 23% depending on population characteristics, diagnostic criteria, and level of access to highly qualified obstetric care [4,5,6]. Maternal mortality in preeclamptic pregnancies is almost four times higher than in pregnancies without preeclampsia [4]. At the same time, perinatal mortality in PE ranges from 10.0 to 30.0% [7,8], which is 2.7 times higher than in pregnancies that lead to full-term birth [7]. PE is not only a cause of maternal and perinatal mortality but also leads to a decrease in the quality of the subsequent life of a woman and her offspring. Women with preeclampsia have an increased risk of developing cardiovascular disease later in life [3]. Previous epidemiological studies indicate a higher incidence of PE in women with gestational diabetes mellitus (7.3%) compared with the general population (4.5%) [9,10,11].

According to current clinical guidelines, women are at greater risk of developing preeclampsia if they have a history of PE, multiple pregnancies, chronic kidney disease, diabetes mellitus, systemic lupus erythematosus, antiphospholipid syndrome, and obesity. Among the most common and significant risk factors for PE are CHTN and obesity. CHTN is defined as an increase in blood pressure ≥140/90 mm hg before pregnancy or during the first 20 weeks of pregnancy, which usually persists for more than 42 days after delivery. However, these factors themselves do not cause the development of preeclampsia, but only modulate the risk of developing the disease. If preeclampsia is overlaid with a pre-existing CHTN, it is called superimposed preeclampsia. Women with superimposed preeclampsia have a higher risk of developing preterm delivery, operative delivery, birth of small-for-gestational-age neonates, admission to the neonatal intensive care unit, and pulmonary edema [12,13,14,15,16].

It is recognized that genetic variations in both the mother and the fetus increase the risk of developing PE, which has a heritability of up to ~55% [17]. There are several studies on molecular genetic risk factors for PE that have been conducted using case-control studies and genome-wide association studies (GWASs). Consequently, more than 100 potential genetic variants linked to this pregnancy pathology have been found [18], one of which is the rs259983 of the Zinc Finger Protein 831 (ZNF831) gene [19,20].

ZNF831 is a gene that encodes a protein and includes nucleic acid binding as part of its function. The gene is located on chromosome 20q13.32. The ZNF831 protein is found in different tissues and organs, including the whole blood, vascular endothelium, pancreas, and placenta. The function of the protein product of the ZNF831 gene has not been thoroughly studied. However, GWAS results in the GWAS-catalog database have demonstrated that the majority of polymorphisms (98 associations within 83 studies) of the ZNF831 gene could affect the course of pregnancy. Therefore, ZNF831 is a promising gene for research in the context of early prenatal diagnosis [19].

A meta-analysis by Steinthorsdottir et al. (2020) made it possible to confirm the association of allele C of the rs259983 polymorphism of the ZNF831 gene with the risk of developing preeclampsia. This study analyzed the genomes of European and Central Asian mothers. Authors found that the variant rs6015450 of the ZNF831 gene had the highest effect on all three blood pressure (BP) traits, especially diastolic BP, which is in strong LD with rs259983 of the ZNF831 gene [20].

In our study, we focused on women with gestational diabetes mellitus (GDM), as preeclampsia often occurs in the context of this condition. The rs259983 polymorphism is not related to the risk of developing diabetes. Therefore, our goal was to examine the relationship between rs259983 of the ZNF831 gene and preeclampsia in pregnant women with GDM. Our second goal was to study the association of polymorphism with preeclampsia in women without CHTN and separately with superimposed preeclampsia.

## 2. Results

### 2.1. Clinical Characteristics

The patient samples were split into two groups: 207 pregnant women without preeclampsia (Control) and 216 patients with PE. For additional analysis, the PE group was subdivided into women with preeclampsia who were not diagnosed with hypertension before pregnancy (PE+, n = 147) and women with preeclampsia superimposed on hypertension (SIPE, n = 69). All participants had GDM. Table 1 summarizes the main clinical characteristics of our study population. There were no significant differences (*p* = 0.075) in age between pregnant women in the PE+ and control groups. Age was statistically significantly different only between SIPE and Control (33.39 ± 4.74 versus 31.30 ± 4.9, *p* = 0.002), but the number of women over the age of 35 was not statistically different. However, we found statistically significant differences in key risk factors for preeclampsia: body mass index (BMI) > 29.9, CHTN, and iron deficiency anemia (IDA). We discovered that pregnant women in the PE+ group had a higher BMI compared to the control group (31.87 ± 8.41 against 26.56 ± 6.68, *p* < 0.00001), and the proportion of clinically obese women was 2.13 times higher. The number of women with CHTN was eight times higher (*p* = 0.00002) among pregnant women with preeclampsia. There were 1.29 times more women with IDA in the PE+ group (*p* = 0.01). When comparing the PE+ and SIPE subgroups with the control group, statistically significant results were found only for SIPE and the Control (87% versus 55%, *p* < 0.00001). GDM treated with insulin is 1.75 times more frequent in the PE group than GDM treated with diet compared to Control (*p* < 0.00001). PE develops 1.64 times more often in pregnant women with GDM who received insulin compared to those treated with diet. There were no statistically significant differences between gestational age at detection of GDM and PE.

We analyzed the interaction between different risk factors such as clinical obesity (which is further defined as a BMI > 29.9), CHTN, IDA, and preeclampsia using multinomial logistic regression (Table 2, Figure 1). All models, except ones adjusted by the IDA Model 4, turned out statistically significant. The most suitable model was model 3, which had the lowest value of the Akaike information criterion (AIC = 513.7). Model 3 when adjusted for obesity, and CHTN, also displayed the greatest sensitivity (62.5) and specificity (71.01). However, these models do not have sufficient specificity (≥80% for clinical tests) and apparently do not include other risk factors for preeclampsia, some of which may not yet be known or poorly understood.

### 2.2. Association between rs259983 of the ZNF831 Gene and Preeclampsia

Before studying the potential relation between rs259983 of the ZNF831 gene and preeclampsia, we calculated the frequency of genotypes and allele occurrences (Table 3). The distribution of genotype and allele frequencies of the rs259983 of the ZNF831 gene matched the one expected in the Hardy–Weinberg equilibrium, both for the PE and for the control group (Control). The C allele frequency in study groups (0.18) was close to the prevalence of polymorphism for the Total (0.25) and European (0.16) populations, according to ALFA Allele Frequency (Release Version: 20230706150541).

The potential connection between rs259983 of the ZNF831 and preeclampsia was then examined (Table 4). No statistically significant association of rs259983 of the ZNF831 gene with preeclampsia was found in any inheritance model. In the codominant inheritance model, the OR reached 1.88 for the CC genotype, but without statistical significance.

### 2.3. Association between rs259983 of the ZNF831 Gene and Preeclampsia without CHTN

According to UKBiobank TOPMed-imputed PheWeb, the rs259983 gene ZNF831 is involved with the following traits: hypertension complicating pregnancy, childbirth and the puerperium [21], essential hypertension and hypertension [22], and calcium channel blocker use measurement and systolic blood pressure [23]. The proportion of pregnant women with SIPE was 31.94%. We suggested that superimposed preeclampsia may be more strongly related to rs259983 of the ZNF831 gene. Therefore, PE groups were split into women with PE, which did not have CHTN (PE+), and women with PE superimposed on CHTN (SIPE) for additional analysis.

For test association only in the PE+ (n = 147) group, we also excluded women with CHTN from the control (n = 198) group. The association with preeclampsia is also absent in all inheritance models (Appendix A). Obesity is the second important risk factor for preeclampsia development, so we studied nonobese women without CHTN in both groups. As a result of our study, no connection with rs259983 of the ZNF831 gene was identified among women without risk factors for preeclampsia (obesity and CHTN) (Appendix A). Therefore, rs259983 does not influence the development of preeclampsia.

### 2.4. Association between rs259983 of the ZNF831 Gene and Superimposed Preeclampsia (SIPE)

HWE was calculated for the control (n = 207) and SIPE (n = 69) subgroups (Table 5). We have found that rs259983 is under HWE in the control group but is not in HWE within SIPE because its *p*-value is < 0.05. For the subgroup, we observed an increase in the frequency of occurrence of the risk allele C by 1.5 times.

The resulting HWE in the control group allowed us to perform an associative analysis (Table 6).

The analysis established the relationship between the rs259983 gene ZNF831 and SIPE in women with GDM. This connection was statistically significant in codominant and recessive inheritance models. The best-fitted model was recessive because it had the lowest AIC value (305.6). In this model, carriers of the CC genotype displayed 5.05 (95% CI: 1.72–14.69) times higher risk of SIPE development.

We additionally performed an association analysis between genes with the risk of developing SIPE to PE+. Statistically significant associations were found in codominant, recessive, and log-additive models (Appendix A).

Thus, the association analysis in codominant and recessive inheritance models has confirmed the role of the risk genotype CC and allele C of polymorphism rs259983 of the ZNF831 in relation to the risk of developing SIPE in women with GDM.

## 3. Discussion

To better understand the contribution of molecular polygenic risk factors to the development of preeclampsia, we studied the relationship between rs259983 of the ZNF831 gene and this pathology in pregnant women with GDM. The results of our study suggest that women with clinical obesity (OR = 2.6, CI 95% = 1.69–4.03) and CHTN (OR = 7.72, CI 95% = 3.85–17.23) have increased odds of developing PE. This logistic model was statistically significant (*p* < 0.0001), but the sensitivity and specificity values indicate that this model is not good enough to use in clinical trials (specificity = 71.01, sensitivity = 62.50, AUC = 70.034).

We did not detect a connection between rs259983 of the ZNF831 gene and PE. However, a relationship was discovered between rs259983 of the ZNF831 gene and SIPE instead. SIPE develops more often in carriers of the AC and CC genotypes (*p* = 0.003). The probability of developing SIPE increased for carriers of the CC genotype by 5.05 times.

### 3.1. Discussion of the PE Risk Factors

The individual risk of developing preeclampsia is modified by such predisposing factors as maternal age under 16 years or over 40 years, nulliparity, multiple pregnancies, increased body mass index before pregnancy, and others. However, individually, none of the risk factors for PE have sufficient power to predict the development of pathology, and even in combination, their predictive power is weak [24]. According to clinical guidelines adopted in Russia, the American College of Obstetricians and Gynecologists (ACOG), and the International Society for the Study of Hypertension in Pregnancy (ISSHP) obesity is considered one of the most significant risk factors for the development of preeclampsia (BMI ≥ 35 kg/m^2^). Literary data also indicate that obesity is a risk factor for the development of preeclampsia [25,26,27]. Thus, Robillard et al., 2019, found that a high BMI before pregnancy was an independent risk factor for the development of preeclampsia. Moreover, an increase in BMI was related only to late preeclampsia, while an increase in age and BMI by 5 years/kg/m^2^ linearly increased the risk of early and late preeclampsia [25]. In a study by Lewandowska et al., 2020, the likelihood of developing preeclampsia is stated to increase in women with a BMI ≥ 30 kg/m^2^ [26]. A retrospective cohort study by Gong et al., 2022, also has shown a 5.06-fold increased risk of preeclampsia in obese women. The authors also noted that the risk of developing preeclampsia is influenced not only by pre-pregnancy BMI, but also by weight gain during pregnancy [27]. These results highlight the key role of obesity in the development of PE, which is consistent with our results.

CHTN is associated with a five-fold increased risk of preeclampsia development compared with normotension [13,28]. The risk of developing SIPE is estimated at 20–50% [14,29,30]. The data vary in different studies, which may be due to local diagnostic criteria for hypertension, characteristics, and organization of medical care. Ellen W. Seely and Jeffrey Ecker, 2014, emphasize that it is important to distinguish between chronic hypertension and new-onset hypertensive complications of pregnancy, such as preeclampsia and gestational hypertension, since the treatment of these diseases differs [31]. According to data from Kametas, N.A. et al., 2022, 20% of pregnant women with chronic hypertension develop PE [16]. In another study, 30% of women with chronic hypertension develop PE [32]. A large proportion of patients with superimposed PE was identified in a 1986 study by Sibai BM et al. where 23 of 44 patients (52%) developed superimposed preeclampsia [30]. In women with end-organ disease or secondary hypertension, the incidence of superimposed preeclampsia has been reported to be as high as 75% [33,34,35,36]. Importantly, treating even mild cases of CHTN with antihypertensive medication before or during the early stages of pregnancy lowers the risk of PE by 18% [37]. In our study, the number of women with CHTN was eight times higher (*p* = 0.00002) among pregnant women with PE. This is also consistent with the literary data.

It is important to note that the presence of CHTN in itself before pregnancy does not cause preeclampsia, nor does the previously mentioned obesity. In actual clinical practice, we rarely encounter isolated cases of diseases; most often, women in labor have a combination of various pathologies, especially obesity and CHTN, the spread of which has increased in recent years. This notion, however, is relevant to the study of genetic predisposition. Most polygenic diseases are associated with different genes, and a single marker can increase the risk of developing various pathologies, especially in the presence of similar molecular and pathophysiological mechanisms. This reasoning can be applied to preeclampsia and CHTN, both of which relate to hypertensive diseases.

### 3.2. Association between rs259983 of the ZNF831 with PE and SIPE

During the analysis, we did not detect an association between rs259983 of the ZNF831 gene and PE in any inheritance model. We obtained similar results by examining the relationship between rs259983 of the ZNF831 gene and preeclampsia, excluding obese women and CHTN from the analysis. These data were not completely consistent with two previous studies, which are GWAS and metaGWAS [18,19]. The key feature of these studies was the search for connections between genotype and phenotype when analyzing thousands and tens of thousands of samples. The large sample size provides high statistical power, but performing analysis on such numerous samples has a number of limitations. The most significant limitation is the heterogeneity of the recruited volunteers. In the GWAS study by Steinthorsdottir et al., 2020, the volunteers/sample was not subdivided into women with PE and preeclampsia superimposed on chronic hypertension. This study did not take into account the influence of risk factors on the development of PE. The sample was not stratified based on age, obesity, chronic hypertension, or other risk factors. However, researchers found that the four PE-associated variants at MECOM, FGF5, ZNF831, and SH2B3 were among the variants with the highest effect on all three blood pressure (BP) traits, particularly diastolic BP. We hypothesized that the identified association was due to the presence of the wide range of pregnant women with SIPE included in this study [19]. Another study found that rs201454025 of the ZNF831 gene is associated not only with PE but also with other maternal hypertensive disorders. At the same time, another polymorphism rs6026744 of the ZNF831 gene within the framework of the same study was only related to PE [38].

Based on meta-analyses of Icelandic and UK Biobank (UKBB) data, Steinthorsdottir et al., 2020, examined the interaction between previously identified preeclampsia-associated polymorphisms and their effects on non-pregnancy diastolic and systolic blood pressure and hypertension. The authors have found that the variations with the largest effects on all three BP traits, especially diastolic BP, and the four preeclampsia-associated variations at rs419076 of the MECOM, rs16998073 of the FGF5, rs3184504 of the SH2B3, and rs6015450 of the ZNF831, and rs6015450 of the ZNF831 are in Linkage Disequilibrium (r^2^  =  0.74 in European), so they all may impact on the pregestation BP and, therefore, could be a risk factor for that trait and not PE as an independent pathology [19]. According to UKBiobank TOPMed-imputed PheWeb, the rs259983 gene ZNF831 is associated with the following traits: hypertension complicating pregnancy, childbirth and the puerperium [21], and essential hypertension and hypertension [22]. With the usage of GWAS Data from FinnGen and UK Biobank on 24 pregnancy complications, Changalidis, A. I. et al. (2022) found loci reaching genome-wide significance in a meta-analysis, including intron variant rs259983 of the gene ZNF831, which shows significant connection to hypertension disorders (meta *p*-value = 8.7 × 10^−7^) [21].

Since the above-mentioned studies emphasized the relationship between rs259983 of the ZNF831 gene and hypertension, we suggested that this polymorphism may have an effect not just on hypertensive disorders during pregnancy. In our study, rs259983 of the ZNF831 gene turned out to be unrelated and not associated with preeclampsia, so we investigated whether this polymorphism could be associated with SIPE instead. We found a statistically significant association between the rs259983 gene ZNF831 and SIPE in women with GDM, which is consistent with the data of Changalidis, A. I. et al. (2022) [21], Steinthorsdottir et al. (2020) [19], and PheWeb data [29]. In our study, carriers of the homozygous CC genotype display an increase in the risk of its development by 5.05 times. The advantages of the study include the fact that all participants were recruited, followed, and assessed at a single facility throughout pregnancy. The novelty of the research is also significant as we provide the first case-control study that estimates the relationship between rs259983 of the ZNF831 gene, preeclampsia, and SIPE in pregnant women with GDM. Moreover, the role of polymorphism in the development of hypertensive complications is yet to be fully investigated and we are ready to offer a PCR test system developed in our laboratory Taq-Man for these exact purposes.

The main limitations of this study are the small sample sizes of the studied groups, which hinder the superimposition of results onto a wider population. For future studies, it is important to consider that a larger sample size will provide more comparable data to the population and increase statistical relevance. In addition, the study included only patients with gestational diabetes without including healthy pregnant women.

Despite limitations that may have influenced the results of the study, the findings indicate the need for further study of the association of the rs259983 polymorphism of the ZNF831 gene to hypertensive complications during pregnancy, as well as with CHTN, with parallel investigation of other genetic variants of the ZNF831 gene that may also affect the development of this pathology. This will make it possible to better assess the contribution of polymorphic variants of the ZNF831 gene to the pathogenesis of these pathologies in representatives of the Caucasian population of the Russian Federation. This study focused on pregnant women with GDM who are at increased risk of preeclampsia in a particular population, and the results of our study can potentially serve as a basis for future research in this area.

## 4. Materials and Methods

### 4.1. Ethical Statement, Exclusion/Inclusion Criteria and Study Design

The study involved DNA samples extracted from the whole venous blood of 423 GDM pregnant women, who were followed up and gave birth in 2019/2022 in the Maternity Department of the State Clinical Hospital No. 29 (N.E. Bauman Hospital) of the Healthcare Department of Moscow. All respondents were Russian speakers of unspecified ethnicity (because of the ethical standards of the local medical register). Each participant gave written and informed consent in accordance with the Helsinki Declaration. The study was also approved by the Ethical Committee of the Research and Development Institute of General Pathology and Pathophysiology.

For the current study participants were split into two groups: 207 pregnant women without preeclampsia (Control) and 216 patients with PE (PE). At the analysis stage, the PE group was divided into women with PE who did not have a CHTN (PE+, n = 147) and women with PE superimposed on CHTN (SIPE, n = 69).

Medical data were extracted from the medical records.

The diagnosis of GDM was established based on the results of the oral glucose tolerance test (OGTT), according to the recommendations of the IADPSG and the criteria of the Russian National Consensus, clinical recommendations, “Gestational diabetes mellitus: diagnosis, treatment, postpartum follow-up” [39,40].

The diagnosis of CHTN, preeclampsia, and superimposed preeclampsia was established on the basis of the clinical guidelines, “Hypertensive disorders during pregnancy, childbirth and the postpartum period. Preeclampsia. Eclampsia” [3].

Pregnant women with a history of autoimmune, cancer, or inflammatory disease were excluded from the study. Women with diagnoses of type 1 diabetes and type 2 diabetes were also excluded.

Minimum sample size was calculated using the quantitative software QUANTO (version 1.2.4, https://bio.tools/QUANTO, accessed on 23 March 2022), which takes into account the frequency of SNPs in the population and the prevalence of the disease. A minimum statistical power of 80% was used in calculating the sample size [41]. Based on the above parameters, a minimum sample size of 174 case-control pairs is required to identify the association between selected polymorphisms and the risk of PE. Taking into account the calculations, 207 pregnant women without preeclampsia and 216 patients with PE were included in the study. All participants have GDM.

Additional information about blood sample collection, transportation, and storage can be found in Appendix A.

### 4.2. Primer and Probe Design

Flanking sequences for rs259983 were obtained using the genomic Ensembl browser [42], https://www.ensembl.org/index.html (accessed on 15 February 2023) in FASTA format (Appendix A).

Primers and probes for genotyping the rs259983 polymorphism of the ZNF831 gene were developed in the online open-access program OligoArchitect Online (http://www.oligoarchitect.com (accessed on 15 February 2023) from Sigma-Aldrich Taufkirchen, Germany) in Dual-Labeled Probe mode. The resulting sequences of primers and TaqMan probes are presented in Table 7.

The In-Silico PCR program at UCSC Genome Browser ( [43], https://genome.ucsc.edu/cgi-bin/hgPcr, accessed on 15 February 2023) was used to test primer specificity and expected amplicon size. All parameters were used by default, taking into account the alignment to the human genome (GRCh38/ch38).

### 4.3. PCR with TaqMan Probes Conditions for rs259983 of the ZNF831

Primers and TaqMan probes were synthesized by Evrogen LLC, Russia.

Amplification was carried out in a programmable thermal cycler Real-Time CFX 96 Touch (Bio-Rad, Hercules, CA, USA). The reaction mixture for RT-PCR for one sample with a volume of 25 μL contained 20–50 ng of DNA, 1X qPCRmix-HS (Evrogen LLC, Moscow, Russia), 200 nM forward primer, 200 nM reverse primer, 250 nM each of the TaqMan probe.

PCR conditions were as follows: initial denaturation for 5 min at 95 °C to activate Hot Start Taq polymerase; then 40 cycles that included 30 s at 95 °C, 30 s at 60 °C and 30 s at 72 °C followed by fluorescence reading. The obtained data were analyzed using CFX Manager TM 3.0 software (Bio-Rad). To eliminate genotyping errors, 30% of randomly selected samples were re-genotyped, and the results were further evaluated. We did not find any errors in repeated genotyping.

### 4.4. Statistical and Computational Analysis

Statistical analysis was performed using SPSS 17.0 (SPSS, Chicago, IL, USA), R 4.0.4 (R Foundation for Statistical Computing, Vienna, Austria), and GraphPad Prism 10 software (“GraphPad”, LLC, San Diego, CA, United States). Visualization was performed using GraphPad Prism 10 software (“GraphPad”, LLC).

To select the correct statistical test, we assessed the normality of data distribution and homogeneity. The normality of the data distribution was checked using the Shapiro–Wilk test. Levene’s Test was used for equality of variance (Homogeneity) between two compared groups. Data are presented as mean ± standard deviation (SD) and using counts with percentages. If the normality of data distribution and homogeneity were met, we used the two-tailed *t*-test; otherwise, we used the Mann–Whitney test. Two-tailed *t*-test or the Mann–Whitney test and χ2 were used to compare parameters between the two groups. The level of significance was considered significant at *p* < 0.05.

Data about polymorphism, associated with preeclampsia and hypertension disorders in the ZNF831 gene, were obtained from NHGRI-EBI GWAS Catalog (www.ebi.ac.uk/gwas, accessed on 10 February 2024) [19]. All SNPs were mapped to Genome Assembly GRCh38.p14 and dbSNP Build 156 [44]. Testing for deviation from Hardy–Weinberg expectations (HWE) was performed using the SNPassoc R package for both cases and controls separately before association analysis. It was considered that the SNP could be analyzed further if there were no significant deviations from HWE in the control group (*p* > 0.05). Deviations from HWE in the case group can indicate issues such as the presence of an association of the analyzed polymorphism with the studied trait. The selection of the best-fitting genetic model was based on the Akaike information criterion (AIC), whereby the best genetic models were those with the lowest AIC values. *p* < 0.05 was considered to indicate a statistically significant association [45].

Logistic regression analysis was performed using GraphPad Prism 10 software (“GraphPad”, LLC) to evaluate the interaction between risk factors and PE by calculating Odds Ratios (ORs) and their 95% Confidence Intervals (CIs). A risk factor was considered significant for pathology if the *p*-value for the effect of the factor was less than 0.05.

## 5. Conclusions

Our study did not find a significant association between the rs259983 variant of the ZNF831 gene and the risk of preeclampsia. The main risk factors for PE development were CHTN and clinical obesity in women with GDM. The sensitivity and specificity of our logistic regression model were not high enough to confidently use it in clinical trials for predicting preeclampsia in these patients, but it underlines the contribution of these risk factors. An association was found between carriers of the AC and CC genotypes rs259983 of the ZNF831 gene and superimposed preeclampsia. The probability of developing SIPE increases for carriers of the CC genotype by 5.05 times. The allele C of the rs259983 polymorphism of the ZNF831 gene increases the risk of developing SIPE, which in turn increases the risk of developing preeclampsia by 1.7 times. These results were obtained for a specific sample, and it is important to conduct additional studies on a wider population in order to confirm our hypothesis.

## Figures and Tables

**Figure 1 ijms-25-11108-f001:**
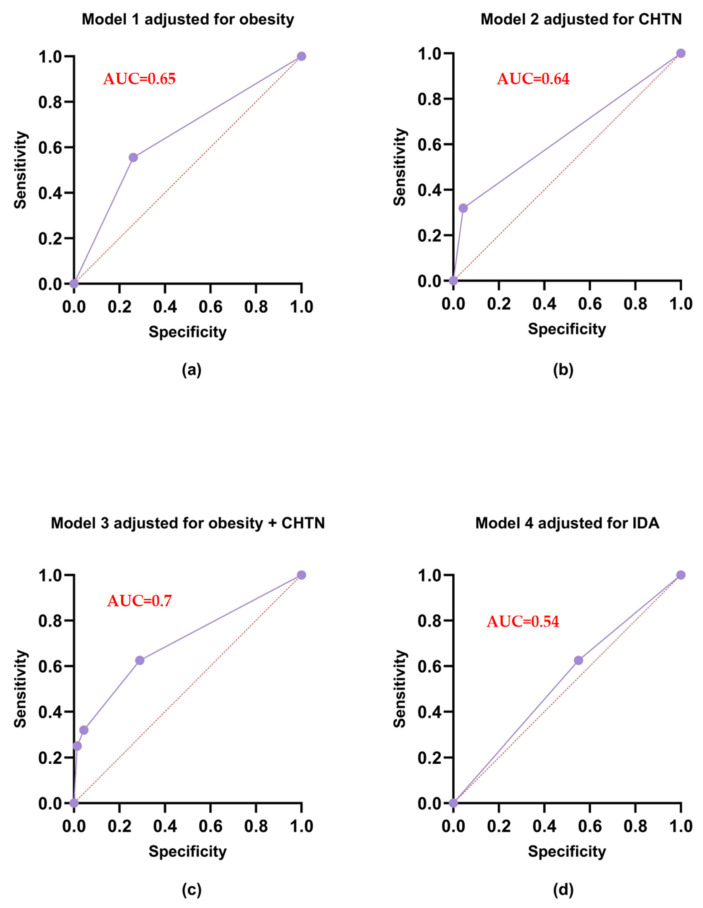
ROC curves for multiple logistic regression models: (**a**) Model 1 adjusted for obesity; (**b**) Model 2 adjusted for CHTN; (**c**) Model 3 adjusted for obesity and CHTN; (**d**) Model 4 adjusted for IDA; AUC (Area Under the Curve): AUC is a single scalar value that summarizes the performance of a binary classifier across all classification thresholds. It represents the probability that a randomly chosen positive instance is ranked higher than a randomly chosen negative instance. An AUC of 0.5 indicates no discrimination (similar to random guessing), while an AUC of 1.0 indicates perfect discrimination; The dots on the ROC curve represent different threshold values used to classify the predictions into positive and negative classes. Each violet dot corresponds to a specific sensitivity (true-positive rate) and specificity (false-positive rate) for that threshold. The straight diagonal pink line on the ROC curve represents the performance of a random classifier. It serves as a baseline for comparison; any classifier that performs better than this diagonal line has some discriminative power, while a classifier that falls below it performs worse than random guessing.

**Table 1 ijms-25-11108-t001:** Comparison of characteristics of study groups between PE and control groups.

Characteristic	Control, n = 207	PE, n = 216		PE+, n = 147		SIPE, n = 69	
	Values	Values	*p*-Value ^1^	Values	*p*-Value ^1^	Values	*p*-Value ^1^
Age (mean ± SD), years	31.30 ± 4.9	32.14 ± 4.8	0.08	31.6 ± 4.6	0.48	33.39 ± 4.74	0.002
≥35 years, n (%)	66 (32)	69 (32)	0.99	39 (26.5)	0.28	30 (43.48)	0.08
BMI (mean ± SD), kg/m^2^	26.6 ± 6.7	31.9 ± 8.4	<0.00001	28.88 ± 6.8	0.0003	38.77 ± 7.83	<0.00001
BMI > 29.9, n (%)	54 (26)	120 (56)	<0.00001	66 (45)	0.0002	54 (78)	<0.00001
CHTN, n (%)	9 (4)	69 (32)	<0.00001	147 (0)	Not applicable ^2^	69 (100)	Not applicable ^2^
IDA, n (%)	105 (55)	135 (63)	0.01	75 (51)	0.96	60 (87)	<0.00001
Insulin/diet GDM treated, n (%)	69 (33)/138 (67)	126 (58)/90 (42)	<0.00001	87 (59)/60 (41)	<0.00001	39 (57)/30 (43)	0.0006
Gestational age at detection of GDM	24.79 ± 7.43	23.37 ± 7.51	0.08	23.15 ± 8.25	0.17	23.83 ± 5.7	0.13
Gestational age at detection of PE	Not applicable ^2^	32.83 ± 4.67	Not applicable ^2^	33.9 ± 3.88	Not applicable ^2^	31.83 ± 5.14	Not applicable ^2^

^1^ Statistically significant result with *p*-value < 0.05, when comparing the group with the Control. The table uses the following abbreviations: PE—preeclampsia; BMI—body mass index; CHTN—chronic hypertension; IDA—iron deficiency anemia; GDM—gestational diabetes mellitus; ^2^ The statistical test cannot be performed because one of the values is 0.

**Table 2 ijms-25-11108-t002:** The multiple logistic regression for PE in pregnant women with GDM.

Model	Variable	OR (95% CI)	*p*-Value ^1^	Sensitivity	Specificity	AIC ^2^
Model 1 adjusted for obesity	Intercept	0.628 (0.485–0.808)	0.0003	55.56	73.91	551.6
	Obesity	3.542 (2.361–5.367)	<0.0001			
Model 2 adjusted for CHTN	Intercept	0.742 (0.599–0.918)	0.006	31.94	95.65	530.5
	CHTN	10.33 (5.245–22.80)	<0.0001			
Model 3 adjusted for obesity and CHTN	Intercept	0.531 (0.405–0.693)	<0.0001	62.50	71.01	513.7
	Obesity	2.601 (1.687–4.034)	<0.0001			
	CHTN	7.720 (3.847–17.226)	<0.0001			
Model 4 adjusted for IDA	Intercept	0.871 (0.646–1.173)	0.36	62.50	44.93	530.5
	IDA	1.36 (0.923–2.001)	0.12			

^1^ Statistically significant result with *p*-value < 0.05. ^2^ AIC—The Akaike information criterion is a mathematical method for evaluating how well a model fits the data it was generated from. A lower AIC indicates a better-fit model.

**Table 3 ijms-25-11108-t003:** Genotype distribution, allele frequency, and *p*-value of Hardy–Weinberg equilibrium for PE and control groups.

Sample Type	Genotype Distribution, n (%)			Allele Frequency		*p*-Value
	AA	AC	CC	A	C	
Control, n = 207	138 (66.67)	63 (30.43)	6 (2.90)	0.82	0.18	0.82
PE+, n = 216	147 (68.06)	57 (26.39)	12 (5.56)	0.81	0.19	0.07

**Table 4 ijms-25-11108-t004:** Association of the rs259983 polymorphism of the ZNF831 gene with PE in pregnant women with GDM.

Model of Inheritance	Genotypes	PE+, n = 216	Control, n = 207	OR (95% of CI)	*p*-Value ^1^	AIC
Codominant	AA	147 (68.1)	138 (66.7)	1.00	0.296	589.8
	AC	57 (26.4)	63 (30.4)	0.85 (0.55–1.30)		
	CC	12 (5.6)	6 (2.9)	1.88 (0.69–5.14)		
Dominant	AA	147 (68.1)	138 (66.7)	1.00	0.761	590.1
	AC+CC	63 (31.9)	69 (33.3)			
Recessive	AA+AC	204 (94.4)	201 (97.1)	1.00	0.172	588.4
	CC	12 (5.6)	6 (2.9)			
Overdominant	AA+CC	159 (73.6)	144 (69.6)	1.00	0.356	589.4
	AC	59 (26.4)	63 (30.4)			
Log-additive	0, 1, 2	216 (51.1)	207(48.9)	1.04 (0.74–1.46)	0.817	590.2

^1^ Statistically significant result with *p*-value < 0.05.

**Table 5 ijms-25-11108-t005:** Genotype distribution, allele frequency, and *p*-value of Hardy–Weinberg equilibrium for SIPE and control groups.

Sample Type	Genotype Distribution, n (%)			Allele Frequency		*p*-Value
	AA	AC	CC	A	C	
Control, n = 207	138 (66.67)	63 (30.43)	6 (2.90)	0.82	0.18	0.82
SIPE, n = 67	42 (60.9)	18 (26)	9 (13.1)	0.74	0.26	0.01

**Table 6 ijms-25-11108-t006:** Association of the rs259983 polymorphism of the ZNF831 gene with superimposed preeclampsia in pregnant women with GDM.

Model of Inheritance	Genotypes	SIPE, n = 69	Control, n = 207	OR (95% of CI)	*p*-Value ^1^	AIC
Codominant	AA	42 (60.9)	138 (66.67)	1.00	0.012	307.67
	AC	18 (26.1)	63 (30.34)	0.94 (0.5–1.76)		
	CC	9 (13.0)	6 (2.9)	4.93(1.66–14.65)		
Dominant	AA	42 (60.9)	138 (66.67)	1.00	0.38	313.7
	AC+CC	27 (39.1)	69 (33.3)	1.29 (0.73–2.26)		
Recessive	AA+AC	60 (87)	201 (97.1)	1.00	0.003	305.6
	CC	9 (13)	6 (2.9)	5.05 (1.72–14.69)		
Overdominant	AA+CC	51 (73.9)	144 (69.6)	1.00	0.49	313.9
	AC	18 (26.1)	63 (30.4)	0.81 (0.44–1.49)		
Log-additive	0,1,2	69 (25)	207 (75)	1.54 (0.99–2.39)	0.057	310.8

^1^ Statistically significant result with *p*-value < 0.05; Statistically significant result highlighted in red font.

**Table 7 ijms-25-11108-t007:** Primer and TaqMan probe sequencing for genotyping the rs259983 of the ZNF831 gene.

Primer Type	Sequences (5’-->3’)	Tm (Salt Adjusted), °C ^1^
Forward primer	GAGGAAGGATGTGGCGAGG	61.6
Reverse primer	GAAGCTGTGGTCAGGAGGAG	62.5
TaqMan probe for A allele	FAM-CTTGTCTCATGG **A** CGCTCTTGATCG-BHQ1	67.4
TaqMan probe for C allele	HEX-CTTGTCTCATGG **C** CGCTCTTGATCG-BHQ1	69.1

^1^ Tm (Salt Adjusted), °C were calculated with; reference and alternative alleles underlined and highlighted in red.

## Data Availability

The original contributions presented in the study are included in the article/Appendix A, further inquiries can be directed to the corresponding author/s.

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
