# Peer review of "Polymorphism rs259983 of the Zinc Finger Protein 831 Gene Increases Risk of Superimposed Preeclampsia in Women with Gestational Diabetes Mellitus"

_ijms, 2024, doi:10.3390/ijms252011108_

Round 1
Reviewer 1 Report
Comments and Suggestions for Authors
This report presents an analysis of the potential association of the rs259983 SNP of the ZNF831 gene with preeclampsia superimposed on chronic hypertension in a sample of pregnancies affected with gestational diabetes.
The following comments are given chronologically with tables and figures moved to the end. Line references are given in [].
[77] The reason for restricting the sample population to only pregnancies diagnosed with GDM is unclear. Is this simply because there is a higher frequency of PE among GDM pregnancies? The authors need to explain their reasoning.
Typically, GDM develops around the 24th week of gestation while PE, by definition, occurs after the 20th week. Thus, it is possible that some pregnancies had PE develop before they developed GDM. Where such pregnancies included in this study? If so then how many?
The authors should include gestational age at detection of GDM and of PE as part of Table 2.
[92-94] “At the analysis stage, the PE group was divided into women with PE who did not have a CHTN (PE+, n=147) and women with PE, which did not have CHTN (PE+) and women with PE superimposed on CHAN (SIPЕ, n=67).” This would be better written: “At the analysis stage, the PE group was divided into women with PE who did not have a CHTN (PE+, n=147) and women with PE and women with PE superimposed on CHTN (SIPЕ, n=67).” A reference to the “Study design” should appear here somewhere.
[104] “… a history of any chronic disease …” This is apart from CHTN, correct?
[106] Including the acronyms “T1DM” and “T2DM” is unnecessary since they are not referenced again within the report. Acronyms are used to shorten future references to a term/phrase.
[107] “Sample size was calculated using …” At this stage the sample size has been determined; calculating and presenting estimates of statistical power is more useful to this presentation.
[164-167] “The Hardy–Weinberg equilibrium (HWE) test was performed with the SNPassoc R package using the chi-square test (χ2) in cases and controls separately before association analysis. It was considered that the SNP could be analyzed further if the p-value in the control group was larger than 0.05.” This would be better written: “Testing for deviation from Hardy–Weinberg expectations (HWE) was performed using the SNPassoc R package for both cases and controls separately before association analysis. It was considered that the SNP could be analyzed further if there were no significant deviations from HWE in the control group (p > 0.05).”
The authors need to explain why they present p-values for the testing for deviations from HWE in the cases when they give no comment of interpretation of these results.
[155-157] The authors give no consequences of the tests for the analysis, i.e., why are they being done.
[157] “absolute numbers” is better written as “counts”
[158] The authors provide a list of statistical tests but do not relate them back to the prior [155-157] distributional tests.
[173-175] “The anticipated risk factor was regarded as significant for pathology if OR adjusted by CI was greater than 1.” No, it is considered significant if the p-value testing for the effect of the factor to be significant is < 0.05. An OR of 1.001 (i.e., greater than 1) would not be significant in this study.
General comment on the Analysis: The authors should have used a logistic regression that included the significant risk factors (obesity and CHTN) as adjustment factors in the genetic models. Thus, an analysis model might be (obesity + CHTN + Dominant). This would be in place of doing separate analysis models within the no CHTN sample, or the no CHTN or obesity sample. As presented it is difficult to determine whether the genetic effect of the ZNF831 gene is separate from a simple CHTN effect.
[181] The term “PE-“ has not been defined.
[188-190] “Additionally, we found that preeclampsia developed 1.75 times more often (p<0.00001) in pregnant women who received Insulin treated GDM.” This is incorrect. GDM treated with insulin is 1.75 times more frequent in the PE group than GDM treated with diet compared to controls. PE developed 1.64 times more often in pregnant women who received Insulin treated GDM compared to those treated with diet. [1.64 = (126/(126+69)) / (90/(90+138))]
[207-208] “Predictive value” has a specific meaning in statistics, especially in the context of diagnostic testing (“positive predictive value”, “negative predictive value”). The authors should rewrite this statement avoiding this term.
[208-209] “… these models do not have sufficient specificity …” How is “sufficient specificity” defined?
[208-209] “However, these models … apparently do not include other risk factors for preeclampsia …” Were (and what) other risk factors were considered in the analysis? If no additional risk factors were considered in the analysis then obviously the models do not include other risk factors. Unclear what the authors mean here.
[292-293] “… however, the sensitivity and specificity indicators do not allow its use in clinical practice” “Allow” is an inappropriately strong a term to use; these sensitivity and specificity values do not prevent use of this as a test, it is just not a very good one.
[
TABLES & FIGURES
[Table 2]
The primary result of the paper is based on a separation of the PE group into PE+ and SIPE, therefore, this table of characteristics should describe these separate groups. A suggestion is columns “Control, n=207”, “PE, n=216”, “PE+, n=149”, “SIPE, n=67”. Omit the “Statistical test” column and add superscripts a, b, c to the p-values to indicate what statistical test was used to obtain the p-value (and include a footnote describing the subscripts).
As noted previously, the authors should include gestational age at detection of GDM and of PE.
Decimal places: Here and throughout the report the number of decimal places is inconsistent and over-extended.
· For percentages, one decimal place (e.g., 31.9%) implies the sample size is at least around 1,000 or more; two decimal places (e.g., 31.94%) implies the sample size is at least 10,000 or more. For sample sizes of 216 or less percentages without decimal places (e.g., 32%) are more appropriate.
· For p-values, two significant digits with a maximum of decimal places are appropriate, unless there are specific journal guidelines stating otherwise. For example: 0.23, 0.023, 0.0023, 0.0002, <0.0001 .
· For measurements the number decimal places depend on the unit of measurement and on the SD. For example, age is (most likely) measured whole years so reporting mean to two decimal places is excessive especially given an SD of 4.76; at most this should be reported as 32.1 ± 4.8 .
“1Statistically significant result with p-value < 0.05” I gather that this explains the odd “< 0.000011” p-values – the superscripts were “desuperscripted”.
[Figure 1]
Given Table 2, Figure 1 does not provide any additional information. Figure 1 should be omitted.
[Table 3]
Were the models compared to determine whether successive models provide a significantly better fit, e.g., Model 1 vs Model 3, Model 2 vs Model 3? Specificity and sensitivity do not provide a test of significantly improved fit of the model.
[Figure 2]
The description for this figure is lacking. The authors do not: define “AUC”, define the dots appearing on the (assumed) ROC curve, the straight diagonal line
[Table 5]
For the Codominant model the authors specify the reference group but do not for Dominant or Recessive models leaving the reader to assume that the reference group is the second group (when it is the first listed group for the codominant). (Why isn’t this table presented in the same manner as Table 7?)
The Overdominant model is confusing. As written it appears to be the same as the Recessive model except with “CC+AC” instead of “AC+CC”. An Overdominant model should be AC / AA+CC (as shown in Table 7!)
The genotype counts for the PE+ do not add up to 216 for the Dominant (sum=210) or Overdominant (sum=218) models.
The authors need to define what the p-values are for.
[Table 7]
“SIPE+” should be “SIPE”
As for Table 5, the authors need to define what the p-values are for.
The authors should include in the supplemental material a table similar to Table 7 that compares SIPE to PE+ (rather than to controls)
[Supplementary Table S5] “Association between rs259983 of the ZNF831 gene and preeclampsia in pregnant women without CHTN and obesity” I suspect this should actually be “ … without CHTN or obesity”.
Comments on the Quality of English Language
The English expression needs careful revision. Some issues are listed among the Comments and suggestions. However, there are other places where the translation from Russian to English is awkward.
I do now know the Russian term for "years"
Author Response
Reviewer 1 Comments
Thank you for such a detailed review of our article. We hope that after the edits, the article has become more understandable and accurate.
General comments on the edits:
All edits were made in a mode that allows you to see the changes. We also corrected the English with a certified translator and an English teacher. Since all the corrections were sent in a separate file, we ourselves included them in the final version of the article.
We also obtained missing data for 2 patients from the SIPE group and recalculated all statistical tests according to the changed data.
We have changed the structure of the article in accordance with the requirements of the journal, changed the numbers of links, tables and images.
Therefore, line numbers may have changed.
Below are comments on each edit, according to your comments.
Comments 1. [77] The reason for restricting the sample population to only pregnancies diagnosed with GDM is unclear. Is this simply because there is a higher frequency of PE among GDM pregnancies? The authors need to explain their reasoning.
Response 1. That’s correct. We added an explanation: “In our study we focused on women with gestational diabetes mellitus (GDM), as preeclampsia often occurs in the context of this condition. The rs259983 polymorphism is not related to the risk of developing diabetes.”
Comments 2. Typically, GDM develops around the 24th week of gestation while PE, by definition, occurs after the 20th week. Thus, it is possible that some pregnancies had PE develop before they developed GDM. Where such pregnancies included in this study? If so then how many?
Response 2. In our study, we specifically focused on pregnancies diagnosed with GDM to ensure a clear understanding of the relationship between GDM and PE. If there were any pregnancies in our sample that had a diagnosis of PE before GDM, they were not included in the final analysis. We do not have specific numbers for such cases, as our criteria aimed to isolate the effects of GDM on the development of PE.
Comments 3. The authors should include gestational age at detection of GDM and of PE as part of Table 2.
Response 3. We included gestational age at detection of GDM and of PE as part of Table 2
Comments 4. [92-94] “At the analysis stage, the PE group was divided into women with PE who did not have a CHTN (PE+, n=147) and women with PE, which did not have CHTN (PE+) and women with PE superimposed on CHAN (SIPЕ, n=67).” This would be better written: “At the analysis stage, the PE group was divided into women with PE who did not have a CHTN (PE+, n=147) and women with PE and women with PE superimposed on CHTN (SIPЕ, n=67).” A reference to the “Study design” should appear here somewhere.
Response 4. We have changed the text according to your recommendations. This proposal already belongs to the section: Ethical statement, exclusion/inclusion criteria and study design.
Comments 5. [104] “… a history of any chronic disease …” This is apart from CHTN, correct?
Response 5. Yes. Corrected the sentence: Pregnant women with a history of autoimmune disease, cancer or inflammatory disease were excluded from the study.
Comments 6. [106] Including the acronyms “T1DM” and “T2DM” is unnecessary since they are not referenced again within the report. Acronyms are used to shorten future references to a term/phrase.
Response 6. We have removed the acronyms
Comments 7. [107] “Sample size was calculated using …” At this stage the sample size has been determined; calculating and presenting estimates of statistical power is more useful to this presentation.
Response 7. Added this information: “Taking into account the calculations, 207 pregnant women without preeclampsia and 216 patients with PE were included in the study. All participants have GDM.”
“A minimum statistical power of 80% was used in calculating the sample size.”
Comments 8. [164-167] “The Hardy–Weinberg equilibrium (HWE) test was performed with the SNPassoc R package using the chi-square test (χ2) in cases and controls separately before association analysis. It was considered that the SNP could be analyzed further if the p-value in the control group was larger than 0.05.” This would be better written: “Testing for deviation from Hardy–Weinberg expectations (HWE) was performed using the SNPassoc R package for both cases and controls separately before association analysis. It was considered that the SNP could be analyzed further if there were no significant deviations from HWE in the control group (p > 0.05).”
Response 8. The sentence has been corrected
Comments 9. The authors need to explain why they present p-values for the testing for deviations from HWE in the cases when they give no comment of interpretation of these results.
Response 9. Added an explanation: Deviations from HWE in case group can indicate issues such as the presence of an association of the analyzed polymorphism with the studied trait
Comments 10. [155-157] The authors give no consequences of the tests for the analysis, i.e., why are they being done.
Response 10. Added an explanation:
“To select the correct statistical test, we assessed the normality of data distribution and homogeneity. The normality of the data distribution was checked using the Shapiro–Wilk test. Levene's Test was used for equality of variance (Homogeneity) between two compared groups. Data are presented as mean±standard deviation (SD) and using counts with percentages. If the normality of data distribution and homogeneity were met, we used the two-tailed T-test, otherwise we used the Mann–Whitney test.Two-tailed T-test or the Mann–Whitney test, and χ2 were used to compare parameters between the two groups. The level of significance was considered significant at p<0.05.”
Comments 11. [157] “absolute numbers” is better written as “counts”
Response 11. The text has been corrected
Comments 12. [158] The authors provide a list of statistical tests but do not relate them back to the prior [155-157] distributional tests.
Response 12. Linked distributional tests to statistical tests: “To select the correct statistical test, we assessed the normality of data distribution and homogeneity. The normality of the data distribution was checked using the Shapiro–Wilk test. Levene's Test was used for equality of variance (Homogeneity) between two compared groups. Data are presented as mean±standard deviation (SD) and using counts with percentages. If the normality of data distribution and homogeneity were met, we used the two-tailed T-test, otherwise we used the Mann–Whitney test.Two-tailed T-test or the Mann–Whitney test, and χ2 were used to compare parameters between the two groups. The level of significance was considered significant at p<0.05.”
Comments 13. [173-175] “The anticipated risk factor was regarded as significant for pathology if OR adjusted by CI was greater than 1.” No, it is considered significant if the p-value testing for the effect of the factor to be significant is < 0.05. An OR of 1.001 (i.e., greater than 1) would not be significant in this study.
Response 13. Correct this sentence: "Logistic regression analysis was performed using GraphPad Prism 10 software (“GraphPad”, LLC) to evaluate the interaction between risk factors and PE by calculating Odds Ratios (ORs) and their 95% Confidence Intervals (CIs). A risk factor was considered significant for pathology if the p-value for the effect of the factor was less than 0.05”
Comments 14. General comment on the Analysis: The authors should have used a logistic regression that included the significant risk factors (obesity and CHTN) as adjustment factors in the genetic models. Thus, an analysis model might be (obesity + CHTN + Dominant). This would be in place of doing separate analysis models within the no CHTN sample, or the no CHTN or obesity sample. As presented it is difficult to determine whether the genetic effect of the ZNF831 gene is separate from a simple CHTN effect.
Response 14. Association analysis was chosen because it allows us to explore complex interactions between genetic factors and health outcomes without having to assume linear relationships, which can be difficult in logistic regression. This is particularly important for identifying the unique influence of SNP such as rs259983 of the ZNF831 on outcomes, independent of the influence of other factors such as obesity and hypertension.
In this study, we analyze a complex trait and the analyzed polymorphism may be associated with preeclampsia indirectly, based on literature and databases.
Since this dependence is nonlinear, it violates the requirements for using regression analysis: The dependence between predictors and results must be linear (the greater one, the greater or lesser the other, without fluctuations);
Predictors must not depend on each other or on the same external factor;
For example: rs259983 increases the risk of developing hypertension, and hypertension increases the risk of developing preeclampsia. In this case, the predictors will depend on each other.
For these reasons, associative analysis in subgroups was chosen.
Comments 15. [181] The term “PE-“ has not been defined.
Response 15. Replaced "PE-" with "Control" used further in the text
Comments 16. [188-190] “Additionally, we found that preeclampsia developed 1.75 times more often (p<0.00001) in pregnant women who received Insulin treated GDM.” This is incorrect. GDM treated with insulin is 1.75 times more frequent in the PE group than GDM treated with diet compared to controls. PE developed 1.64 times more often in pregnant women who received Insulin treated GDM compared to those treated with diet. [1.64 = (126/(126+69)) / (90/(90+138))]
Response 16. Thank you for this comment, we have corrected the text.
Comments 17. [207-208] “Predictive value” has a specific meaning in statistics, especially in the context of diagnostic testing (“positive predictive value”, “negative predictive value”). The authors should rewrite this statement avoiding this term.
Response 17. Removed the term from the sentence "Predictive value": Model 3 adjusted for obesity and CHTN had the greatest sensitivity (62.5) and specificity (71.01).
Comments 18. [208-209] “… these models do not have sufficient specificity …” How is “sufficient specificity” defined?
Response 18. A specificity of ≥ 80% is often considered sufficient in many clinical settings. This means that the model correctly identifies at least 80% of true negatives.
Added this remark: “However, these models do not have sufficient specificity (≥80% for clinical test) and apparently do not include other risk factors for preeclampsia, some of which may not yet be known or poorly understood.”
Comments 19. [208-209] “However, these models … apparently do not include other risk factors for preeclampsia …” Were (and what) other risk factors were considered in the analysis? If no additional risk factors were considered in the analysis then obviously the models do not include other risk factors. Unclear what the authors mean here.
Response 19. Here we have risk factors that are not included in clinical guidelines because they have been poorly studied or do not yet have a sufficient evidence base.
Comments 20. [292-293] “… however, the sensitivity and specificity indicators do not allow its use in clinical practice” “Allow” is an inappropriately strong a term to use; these sensitivity and specificity values do not prevent use of this as a test, it is just not a very good one.
Response 20. Corrected this sentence: This logistic model was statistically significant (p<0.0001), but the sensitivity and specificity values indicate that this model is not good enough to use in clinical trials (specificity=71.01, sensitivity=62.50, AUC=70.034)
TABLES & FIGURES
Comments 21. [Table 2]
The primary result of the paper is based on a separation of the PE group into PE+ and SIPE, therefore, this table of characteristics should describe these separate groups. A suggestion is columns “Control, n=207”, “PE, n=216”, “PE+, n=149”, “SIPE, n=67”. Omit the “Statistical test” column and add superscripts a, b, c to the p-values to indicate what statistical test was used to obtain the p-value (and include a footnote describing the subscripts).
As noted previously, the authors should include gestational age at detection of GDM and of PE.
- For percentages, one decimal place (e.g., 31.9%) implies the sample size is at least around 1,000 or more; two decimal places (e.g., 31.94%) implies the sample size is at least 10,000 or more. For sample sizes of 216 or less percentages without decimal places (e.g., 32%) are more appropriate.
- For p-values, two significant digits with a maximum of decimal places are appropriate, unless there are specific journal guidelines stating otherwise. For example: 0.23, 0.023, 0.0023, 0.0002, <0.0001 .
- For measurements the number decimal places depend on the unit of measurement and on the SD. For example, age is (most likely) measured whole years so reporting mean to two decimal places is excessive especially given an SD of 4.76; at most this should be reported as 32.1 ± 4.8 .
“1Statistically significant result with p-value < 0.05” I gather that this explains the odd “< 0.000011” p-values – the superscripts were “desuperscripted”.
Response 21. We took into account your recommendations on the design of the table and included 4 groups: “Control, n=207”, “PE, n=216”, “PE+, n=149”, “SIPE, n=69”. Since we did not have data on CHTN for 2 patients, we requested them from the hospital. These data were included in the table, and we also recalculated the association analysis. We also calculated the p-value for each group “PE, n=216”, “PE+, n=149”, “SIPE, n=69” in comparison with the control. The table included data on the timing of diagnosis of GDM and preeclampsia.
Decimal places: Here and throughout the report the number of decimal places is inconsistent and over-extended.
Superscripts have been added in the appropriate places.
We have corrected the number of decimal places
We included gestational age at detection of GDM and of PE as part of Table 2
Comments 22. [Figure 1]
Given Table 2, Figure 1 does not provide any additional information. Figure 1 should be omitted.
Response 22. Figure 1 was omitted
Comments 23. [Table 3]
Were the models compared to determine whether successive models provide a significantly better fit, e.g., Model 1 vs Model 3, Model 2 vs Model 3? Specificity and sensitivity do not provide a test of significantly improved fit of the model.
Response 23. We calculated the AIC to compare the models, added the values ​​to the table, and also indicated the best fitted model.
“The most suitable model was model 3, which had the lowest value of The Akaike information criterion (AIC=513.7)”
Comments 24. [Figure 2]
The description for this figure is lacking. The authors do not: define “AUC”, define the dots appearing on the (assumed) ROC curve, the straight diagonal line
Response 24. We have added all missing explanations:
AUC (Area Under the Curve): AUC is a single scalar value that summarizes the performance of a binary classifier across all classification thresholds. It represents the probability that a randomly chosen positive instance is ranked higher than a randomly chosen negative instance. An AUC of 0.5 indicates no discrimination (similar to random guessing), while an AUC of 1.0 indicates perfect discrimination; The dots on the ROC curve typically represent different threshold values used to classify the predictions into positive and negative classes. Each dot corresponds to a specific sensitivity (true positive rate) and specificity (false positive rate) for that threshold; The straight diagonal line on the ROC curve represents the performance of a random classifier. It serves as a baseline for comparison; any classifier that performs better than this diagonal line has some discriminative power, while a classifier that falls below it performs worse than random guessing.
Comments 24. [Table 5]
For the Codominant model the authors specify the reference group but do not for Dominant or Recessive models leaving the reader to assume that the reference group is the second group (when it is the first listed group for the codominant). (Why isn’t this table presented in the same manner as Table 7?)
The Overdominant model is confusing. As written it appears to be the same as the Recessive model except with “CC+AC” instead of “AC+CC”. An Overdominant model should be AC / AA+CC (as shown in Table 7!)
The genotype counts for the PE+ do not add up to 216 for the Dominant (sum=210) or Overdominant (sum=218) models.
The authors need to define what the p-values are for.
Response 24. The authors apologize for the errors in Table 5. We have corrected everything in the same way as in Table 7.
p-values ​​allow you to determine which models are statistically significant. Added explanation below the table: 1 Statistically significant result with p-value < 0.05; Highlighted in red font.
Comments 25. [Table 7]
“SIPE+” should be “SIPE”
Response 25. It has been fixed
Comments 26. As for Table 5, the authors need to define what the p-values are for.
Response 26. The p-values for table 7 are defined: Statistically significant result with p-value < 0.05; Highlighted in red font.
Comments 27. The authors should include in the supplemental material a table similar to Table 7 that compares SIPE to PE+ (rather than to controls)
Response 27. The corresponding calculations have been added to supplemental material
Comments 28. [Supplementary Table S5] “Association between rs259983 of the ZNF831 gene and preeclampsia in pregnant women without CHTN and obesity” I suspect this should actually be “ … without CHTN or obesity”.
Response 28. It has been fixed
Reviewer 2 Report
Comments and Suggestions for Authors
Thanks to the authors for an overall very well structured, interesting study.
Very imporant to make genotyping to a clinically useful issue. Although correlation may be weak, the tendency is an important confirmation.
only few points which could be adjusted and specified:
1. abstract: HDP danger especially in pregnancy...HDP describes hypertension in pregnancy?
2. line 27: mortality would be preventable...is this true for all countries?
Author Response
Thank you for review of our article. We hope that after the edits, the article has become more understandable and accurate.
General comments on the edits:
All edits were made in a mode that allows you to see the changes. We also corrected the English with a certified translator and an English teacher. Since all the corrections were sent in a separate file, we ourselves included them in the final version of the article.
We also obtained missing data for 2 patients from the SIPE group and recalculated all statistical tests according to the changed data.
We have changed the structure of the article in accordance with the requirements of the journal, changed the numbers of links, tables and images.
Therefore, line numbers may have changed.
Below are comments on each edit, according to your comments.
Comments 1. abstract: HDP danger especially in pregnancy...HDP describes hypertension in pregnancy?
Response 1.Yes, it is. This term is widely used in scientific and medical literature, for example: https://www.escardio.org/Journals/E-Journal-of-Cardiology-Practice/Volume-17/hypertension-in-pregnancy
Comments 2. line 27: mortality would be preventable...is this true for all countries?
Response 2. Yes, but it depends on the conditions of providing medical care in different countries, as well as socio-economic aspects in general